# Physical Activity and Self-Reported Metabolic Syndrome Risk Factors in the Aboriginal Population in Perth, Australia, Measured Using an Adaptation of the Global Physical Activity Questionnaire (GPAQ)

**DOI:** 10.3390/ijerph18115969

**Published:** 2021-06-02

**Authors:** Tuguy Esgin, Deborah Hersh, Kevin G. Rowley, Rona Macniven, Kristen Glenister, Alan Crouch, Robert U. Newton

**Affiliations:** 1Discipline of Exercise, Health and Performance, The University of Sydney, Sydney, NSW 2006, Australia; 2School of Medical and Health Sciences, Edith Cowan University, Perth, WA 6027, Australia; d.hersh@ecu.edu.au (D.H.); r.newton@ecu.edu.au (R.U.N.); 3Onemda VicHealth Koori Health Unit, University of Melbourne, Melbourne, VIC 3010, Australia; 4School of Population Health, University of New South Wales, Kensington, NSW 2052, Australia; r.macniven@unsw.edu.au; 5Poche Centre for Indigenous Health, Faculty of Medicine and Health, The University of Sydney, Camperdown, NSW 2006, Australia; 6Faculty of Health, Medicine and Human Sciences, Macquarie University, North Ryde, NSW 2109, Australia; 7Department of Rural Health, University of Melbourne, Shepparton, VIC 3630, Australia; Kristen.glenister@unimelb.edu.au (K.G.); alan.crouch@unimelb.edu.au (A.C.); 8School of Human Movement and Nutrition Sciences, University of Queensland, Brisbane, QLD 4072, Australia

**Keywords:** Aboriginal and Torres Strait Islander, oceanic ancestry group, physical activity, GPAQ

## Abstract

*Background:* Complex, ongoing social factors have led to a context where metabolic syndrome (MetS) is disproportionately high in Aboriginal Australians. MetS is characterised by insulin resistance, abdominal obesity, hypertension, hypertriglyceridemia, high blood-sugar and low HDL-C. This descriptive study aimed to document physical activity levels, including domains and intensity and sedentary behaviour, and MetS risk factors in the Perth Aboriginal (predominately Noongar) community. *Methods:* The Global Physical Activity Questionnaire (GPAQ), together with a questionnaire on self-reported MetS risk factors, was circulated to community members for completion during 2014 (*n* = 129). *Results:* Data were analysed using chi-squared tests. The average (SD) age was 37.8 years (14) and BMI of 31.4 (8.2) kg/m^2^. Occupational, transport-related and leisure-time physical activity (PA) and sedentary intensities were reported across age categories. The median (interquartile range) daily sedentary time was 200 (78, 435), 240 (120, 420) and 180 (60, 300) minutes for the 18–25, 26–44 and 45+ year-olds, respectively (*p* = 0.973). *Conclusions:* An in-depth understanding of the types, frequencies and intensities of PA reported for the Perth Aboriginal community is important to implementing targeted strategies to reduce the prevalence of chronic disease in this context. Future efforts collaborating with community should aim to reduce the risk factors associated with MetS and improve quality of life.

## 1. Introduction

Physical activity is defined as “bodily movement produced by the skeletal muscle that results in energy expenditure” [1] and can be categorised in different domains, typically work, transport and discretionary (leisure or recreation) [2]. Physical activity can be quantified through the metabolic equivalent of task (MET), a physiological measure expressing the energy cost or calories required for different physical activities [3] with one MET being the energy equivalent expended by an individual while seated at rest. Sedentary behaviour is distinct from physical inactivity and is defined as any waking behaviour characterized by an energy expenditure ≤1.5 METs while in a sitting or reclining posture. Increased sedentary lifestyles have an adverse effect on health, leading to several chronic diseases, and are associated with higher mortality rates [4]. In this context, an understanding of population levels of activity is important for the design and development of campaigns that seek to prevent conditions such as metabolic syndrome (MetS). MetS is the clustering of risk factors associated with sedentary lifestyles, including obesity, dyslipidemia, hyperinsulinemia, impaired fasting glucose and hypertension [5] as defined by the US NCEP’s Adult Treatment Panel III [6]. It has been associated with lifestyle patterns including low levels of physical activity. 

Aboriginal people, who are the oldest continuous civilisation in the world, have high rates of MetS and cardiovascular disease (CVD). This has been attributed to the influence of negative historical and social experiences since the time of colonisation [7]. There is an extensive literature- and evidence-base on factors and conditions contributing to the health and wellbeing of Aboriginal and Torres Strait Islander peoples, including national burden-of-disease [8] and health determinants data [9,10,11]. Analyses of these data highlight opportunities for strengthening the duration and quality of life of Aboriginal and Torres Strait Islander people by addressing the causes of avertible morbidity and premature mortality [12]. These data include indications that the gap in life expectancy and the higher hospitalisation rates among Aboriginal Australians are mostly associated with CVD and metabolic disease [13]. The first author, who is a member of the Noongar and Yamatji Aboriginal community, worked closely with the community to develop a questionnaire designed to elicit self-reported MetS risk factors and the motivators and barriers to exercise [14]. 

“Noongar” means ‘a person of the south-west of Western Australia’, (South West Aboriginal Land and Sea Council, 2013) and many Noongar people now live in the capital city of Western Australia, Perth. 

The questionnaire also included the standard Global Physical Activity Questionnaire (GPAQ) that measures physical activity in three settings, or domains, as well as sedentary behaviour. 

A previous study with Indigenous Australian participants in Brisbane initially attempted to use the IPAQ-long survey format (International Physical Activity Questionnaire-long form). However, the use of this instrument was discontinued after participants communicated to research assistants that the questionnaire was not well understood and that it led to participants feeling frustrated with the project [15]. In the Cree (First Nations) community in Canada, the IPAQ-long was considered by community members to be too long and burdensome and a validity study of the IPAQ-short was conducted, following translation into local language and cultural adaptation to include appropriate physical activity examples [16]. The aforementioned study concluded that more work was required to ensure the cultural adaptation of the IPAQ for Indigenous peoples. 

The GPAQ is an enhanced version of the IPAQ aimed at capturing physical activity measured in cross cultural settings; it contains only 16 questions [17]. The GPAQ, though not previously used to measure physical activity among Aboriginal Australians [18], has been adapted for a range of culturally diverse populations [18]. Given its demonstrated use among diverse, international population groups [2], the GPAQ, used in conjunction with the MetS risk factor questionnaire, was selected for this study to elicit data on physical activity levels of Aboriginal adults (predominately of Noongar heritage) on Noongar Whadjuk Boodja country (Perth Metropolitan area, in Western Australia (WA)) [14]. 

This descriptive study aimed to document physical activity levels, including domains and intensity and sedentary behaviour, and MetS risk factors in the Perth Aboriginal (predominately Noongar) community The data obtained were further used to prescribe physical activity that was culturally appropriate and that specifically addressed the major Indigenous health issues around MetS and its risk factors [19].

## 2. Materials and Methods

The philosophy of this decolonising framework emphasised Indigenous research methodologies and the promotion of Indigenous world views. An Indigenous standpoint theory (IST), in which the researcher establishes a ‘positionality’ within their ancestors’ culture and knowledge, forms the epistemological foundation for this study. Methodological choices were made within this theoretical framework to ensure culturally responsive research processes that engaged the Indigenous agenda of self-determination and rights [20,21]. In this context, participatory action research methods were used to embed culturally responsive community co-design and co-production processes across the study [14]. This enquiry is a PAR cross-sectional study of the development, method and survey distribution of the questionnaire for the broader investigation within which this study was nested and which has been described previously [14]. This community consultation process resulted in the addition of examples of physical activity, including fishing and hunting for food, regarded as culturally appropriate, into the GPAQ version for the present study. The questionnaire components for the present study also included an additional questionnaire with self-reporting of risk factors for MetS [14]. 

### 2.1. Community Engagement

Community engagement in the research was facilitated by the first author (TE). The consultation phase with the Perth Aboriginal community, facilitated through Derbarl Yerrigan Aboriginal Medical Service, established the relevance of the GPAQ and resulted in the inclusion of cultural examples of activities such as walking, hunting and fishing. This consultation included gaining the endorsement of local Elders for these additions. A process of participant consultation on the cultural appropriateness of the GPAQ instrument was conducted in the initial phase of the survey implementation by TE. Consequently, the GPAQ questionnaire format was modified to include extra spacing between lines and enlarged font size. The feedback from participants was that the changes increased the readability of the instrument.

### 2.2. Participants

Participants eligible to take part were any Aboriginal person over 18 years old living in Perth, WA. Initial response following email requests to Aboriginal community organisations was low. The first author then drew on extensive community networks and made direct contact with potential participants in local community-controlled organisations and shopping centres to recruit participants.

### 2.3. Measures

The WHO describes the development of a standardized tool (GPAQ) to measure physical activity. The authors have chosen to use the term ‘measures’ to reflect the use of this term by WHO and in other papers using the GPAQ [22]. The GPAQ utilises questions to elicit the quantity and domains of physical activity relating to occupational, transport-related and leisure-time activities, in minutes per day, as well as sedentary behaviour. Participants were asked to report on several physiological and anthropometric measures. These self-reported measures included height and weight (which were then used to calculate body mass index, BMI), blood pressure and type 2 diabetes status. Previous studies have reported that the accuracy of self-reported hypertension and diabetes are acceptable in Australian samples [23,24]. Data related to medication use were not collected.

### 2.4. Procedure

Participation in the study was voluntary and no incentives were offered to the respondents. The study procedure was fully explained to each participant and written informed consent was obtained from all participants prior to administration of the survey. The questionnaire was completed by participants while the first researcher was present, to assist with participants’ questions, and to ensure completion and return of the questionnaire. This style of research with direct contact was culturally appropriate, considering the first author’s history and active involvement in the Noongar community. 

### 2.5. Data Analysis 

In line with the GPAQ Analysis Guide, the responses were converted to MET-minutes per week. Moderate activity is classified as 4 METs and vigorous activity 8 METs [25]. Further, the respondents’ physical activity was classified as high, moderate or low depending on their total MET-minutes per week or other combination criteria [26]

Kolmogorov–Smirnov (K–S) tests indicated that the data were not normally distributed. Chi-squared tests were used to describe the study population and to determine the prevalence of various levels of physical activity and sedentary behaviour in age (18–25, 26–44 and 45+ years) and sex (male, female) groups. First, the proportion of ‘highly active’, ‘moderately active’ and ‘low active’ adult females and males in different age categories were described. Secondly, the median time spent (MET-minutes per day) during a physical activity at work, travel to and from places, recreational physical activity and sedentary behaviour in age and sex groups were described. Sedentary behaviour was classified according to minutes per day of sitting at a desk, sitting with friends, travelling by car, bus, or train, reading, playing cards or watching television, excluding the time spent for sleeping [25].

Physical activity (at work, travel to and from places and recreational physical activity including walking) expressed as MET-minutes per day and sedentary behaviour (expressed as minutes per day) were used as dependent variables in inferential statistical models of the correlates of physical activity. Age categories (18–25, 26–45 and 45+ years), sex (female and male) and risk factor status were used as independent variables. The Kruskal–Wallis test was used to detect differences in median values for different age and sex categories. The generalized linear model was used to estimate the effect of age, sex and self-reported MetS risk factor status in predicting levels of physical activity (Table 5). The assumption of independence was used for this model. The probability distribution of the dependent variable, the intensity of physical activity, was specified as multinomial, in three categories: low, moderate and high.). The data were analysed using IBM SPSS v.22 (IBM Corporation, Armonk, USA), with statistical significance set at *p* ≤ 0.05. Cumulative logit link function used in this analysis is the default for multinomial modeling. f(x) = ln(x/(1 − x))

The dependent variable was sorted in ascending order. The intercept was included in the model. The statistical criterion for the generalized linear model was type III WALD which provided chi-square statistics for each effect. The confidence interval level was specified to be 95% and the confidence interval type was WALD. The covariance matrix was used to estimate the parameters in the model. The model parameters estimation method used was the hybrid Fisher scoring method, allowing for one iteration before switching to the Newton–Raphson method. This method requires convergence for both Fisher and Newton–Raphson. The parameter convergence criterion used was 1 × 10^−6^ (or 1E-006) and the tolerance value for testing singularity was 1 × 10^−12^ (or 1E-012). These measures were calculated with the assistance of a statistics consultant.

Ethics approval for this study was obtained from the Aboriginal Health Council of WA HREC reference number 365-09/11 and Edith Cowan University Ethics Committee project code approval number 5818.

## 3. Results

### 3.1. Acceptability

The Noongar (Aboriginal) researcher for this study observed that the administration of the GPAQ on urban Australian Indigenous people was relatively straight-forward. A total of 129 participants completed the questionnaire, either one-to-one with the researcher or in small groups with the researcher present to assist if they had any trouble. All participants who started the questionnaire completed it. 

### 3.2. Demographic and Clinical Characteristics of Participants

Just under half of the *n* = 129 participants (46.5%) were between 26 and 45 years of age (Table 1). The average age of the participants was 37.8 years (ranging from 18 and 71 years) and the average BMI was 31.1 kg/m^2^ (obese; BMI ≥ 30 kg/m^2^), calculated from self-reported figures on height and weight. Self-reported high blood pressure *n* = 43 (33%) and type 2 diabetes (T2D) *n* = 39 (30%) were the most prevalent MetS risk factors for females, whereas, for males, T2D *n* = 30 (23%) and high cholesterol *n* = 30 (23%) were most prevalent. A quarter of the participants in this study had high blood pressure, T2D and high cholesterol. Overall, 23% of the participants had MetS. Females had higher percentages for MetS risk factors, and almost double the male levels for heart disease (10% vs. 5%) and hypertension (33% vs 18%), Table 1. 

The median time spent by participants in physical activity in different settings, and time spent being sedentary is presented instead of mean data because the data have skewed distributions. The different intensities of physical activity the participants undertook across age categories is presented in Table 2. 

For activity at work, 25 of the participants aged 26–45 years reported a median of 15 min per day, which was the highest moderate activity type. Results were either zero or close to zero for vigorous and moderate for the other two age groups. Travel to and from places accounted for *n* = 32 (25%) of participants aged 18 to 25 years, recording a median of 11 min. Similarly, *n* = 32 (25%) of those aged 26 to 45 and aged 45+ recorded 13 min or 2 min, respectively, higher than the younger group, however, this difference was not statistically significant.

Only vigorous recreational activities varied significantly by age with the 45+ age category showing much lower median times of 0 min at the 25th percentile (0,4) compared with participants aged 26–45 years (0,26) and 18–25-years (0,18). The highest recorded recreational activity was within the 26–45 age category at 26 min per day or higher, followed by the 18–25 age category at 18 min per day or higher and the 45+ age category with 4 min of activity. Moderate activity (recreation activities) saw the top 25% of participants aged 26–45 years and 45+ years equal at 17 min per day or more. For sedentary time, those aged 26–45 years performed the most activity while also recording the most sedentary time. 

Consistent with the GPAQ analysis guide, [25], health benefits are achieved when physical activity levels range between 500 and 1000 MET-minutes per week. Participants in this study achieved the minimum MET-minutes when participating in recreational vigorous activity, 26 min in the 26–45 age category and 18 min in the 18–25 age categories. Some significant differences between sex for the level of activity (MET-minutes per week) are detailed (Table 3). 

The median moderate recreational physical activity MET-minutes per week was significantly higher among males than females (*p* = 0.029). For the total time of all activity, males at 435 MET-minutes per week nearly doubled females at 240 MET-minutes per week. However, this was not found to be significant (*p* = 0.071).

The median sedentary time did not vary by risk factor status for men, women or the whole survey sample. Males without risk factors tended to have a higher median time physically active, but this did not reach statistical significance (*p* < 0.05) for diabetes and high cholesterol. However, people without reported heart disease spent more time being physically active on average (*p* < 0.05). There was no significant difference in median time spent physically active between persons with normal waist circumference and those with elevated waist circumference, for either men or women (Table 4).

Using linear regression, undertaking physical activity of moderate intensity (β = 1.39, *p* < 0.001) and being male (β = 0.82, *p* = 0.015) were predictive of increased levels (expressed as MET-minutes per week) of physical activity, as per Table 5. 

## 4. Discussion

This study has described the application of the internationally validated GPAQ instrument, culturally adapted for use with the Perth Aboriginal community. It was found to be an acceptable and valid tool to measure physical activity and sedentary behaviour in this context. It is believed to be the first study to report use the GPAQ to measure physical activity and sedentary behaviour of Aboriginal and Torres Strait Islander peoples. Internationally, it is the first time the GPAQ has been used with other First Nations peoples (Indigenous) from countries with similar settler-colonial histories to Australia—Canada, the United States and New Zealand. The findings also suggest that the GPAQ could be an appropriate and useful way to measure physical activity and sedentary behaviours among other Aboriginal and Torres Strait Islander peoples and communities in Australia. Given the Canadian [16] and Australian [15] experiences with the IPAQ in Indigenous communities, consideration should be given to adapting the modified GPAQ for use in other international First Nations populations. This finding is strengthened by the overall community engagement with the GPAQ and the inclusion of examples relevant to the Perth Aboriginal community. The importance of a culturally relevant and appropriate tool to gather physical activity data and sedentary behaviour levels in Indigenous communities is critical to the development of programs, projects or interventions that use physical activity is a means to both develop health promotion programs and to improve health outcomes. This is particularly important as a recent review found that few CVD programs designed specifically for Aboriginal Australians have undergone rigorous study and few physical activity programs for Aboriginal Australians have included a thorough evaluation including baseline participant data [27,28]. 

These findings highlight several differences between the types of physical activity according to sex and age among the Perth Aboriginal community. Overall, males were more active than females. Vigorous-intensity physical activity was higher among younger (18–25 years) than older participants which may be due to younger adults being more able to achieve more physically intensive activity levels as cardiorespiratory fitness decreases by 0.245 mL/min/kg per year on average [29]. This study is novel as very few studies conducted with Aboriginal and Torres Strait Islander communities have measured physical activity comprehensively, specifically investigating vigorous physical activity. Males had higher levels of moderate recreational activity than females which may be due to sex-specific preferences for recreational sport. Indeed, increased duration and intensity of exercise were associated with males to a greater extent than females, potentially due to differences in the types of physical activity undertaken. An understanding of sex differences in physical activity is important to inform targeted strategies to improve health outcomes, as sex-specific activities may be preferred. National data on physical activity levels of Aboriginal adults indicate that males are more active than females [30]. Other than a study of Aboriginal adults aged 45 years and over in New South Wales, [31] which did not find sex differences, an understanding of Aboriginal physical activity among adults has been unreported in urban populations, before this study. 

A European GPAQ study [32] reported that men are more engaged in physical activity at work and during travel and leisure-times than women and travel-related METS-minutes were the lowest domain of physical activity for both sexes [32]. In contrast, in our study, work-related MET-minutes were the lowest physical activity domain for both sexes. Travel-related MET-minutes were higher for males than for females and walking has been previously identified as the preferred method of exercise [19]. The 22% higher MET-minutes for females noted in this study may be attributable to walking by females. Why the Aboriginal women in our study walk more than the Aboriginal men is uncertain, but it may be that they prefer walking or that they have fewer other transport options, including access to a private vehicle. 

The median values for all activity for men and women in this study were 41 and 21 min per day, respectively, which roughly corresponds to men, but not women, meeting the lower threshold of the physical activity guidelines of 150 min per week (30 min per day) [33]. A previous study of 314 Aboriginal people aged 45 years and over in New South Wales (NSW), using the Active Australia Survey that has similar physical activity intensity categories, found that the majority, 63%, met the upper threshold of the physical activity guidelines [31]. A potential reason for this difference could be socioeconomic differences in the participant groups; the NSW study included participants of a higher socioeconomic status than the case among Aboriginal people in Australia more generally, but measuring socioeconomic factors was beyond the scope of the present study. 

Many previous studies indicate that sedentary activity accounts for a substantial amount of time in the average Australian’s day but there is little previous evidence of sedentary behaviour levels among Aboriginal Australians, beyond national statistical data [34,35]. Data found, in 2012–2013, that Aboriginal adults in non-remote areas spent an average of 5.3 h per day on sedentary behaviour [36]. In this study, sedentary activity occupied an average of 23.3 h per week (200 min or 3.33 h/day). The median minutes of sedentary time per day for Perth Aboriginal participants were 200, 240 and 180 for those aged 18–25, 26–45 and 45 + years, respectively. 

Sixty per cent of adults across the age categories spent four or more hours a day sedentary; the highest rate of sedentary behaviour was observed in the age category of >45. This corresponds with the European average of 64.1% and the world average is 41.5% (Hallal et al., 2012). The figures obtained from this study reflect those found in Europe, which are almost 20% higher than in the rest of the world. The continuing increase in overweight and obesity is associated with a decrease in the level of physical activity and increases in sedentary behaviours among Australian Aboriginal people [37]. Greater emphasis needs to be placed on reducing sedentary behaviour and increasing physical activity levels of Australian Aboriginal people in culturally relevant ways to improve health and quality of life outcomes [38]. 

There are strong links between physical activity and obesity [39] often measured through BMI, and the average BMI of participants in this study was high at 31 kg/m^2^. The most recent Australian Aboriginal and Torres Strait Islander Health Survey, in 2012–2013, also found high BMI levels with two-thirds (66%) overweight or obese (29% and 37%, respectively) [40]. Programs and strategies that focus on increasing physical activity will likely have a cumulative, twofold impact in both reducing chronic disease risk factors and reducing weight which is also a risk factor for chronic disease. 

Lower physical activity rates in participants aged between 18 and 25 years may relate to indoor office roles or being engaged in education and study that do not require much physical exertion. Another reason for the lower rate of activity in the younger group could be involvement in childcare activities. This was found to be a barrier to exercise in a previous related study and another recent study in NSW that used a similar version of the questionnaire that was developed with the community [41]. This may explain the higher physical activity rates observed in participants aged 26–45 years, whose occupations and leisure activities may have been more physically demanding. 

While risk factors for chronic disease were prevalent among participants, the only significant differences in physical activity levels across disease risk factors were for heart disease. Those with heart disease had lower physical activity levels than those who did not have heart disease. Heart disease was also more common among females than males in these participants. The prevalence of heart disease within Indigenous communities has been documented in studies undertaken since the 1970s. Bastian [42] reported that Aboriginal women from the West Kimberley region of Western Australia similarly had an almost double prevalence of coronary heart disease (11%) than men (7%). In a study in a remote Aboriginal community in the Northern Territory where 920 participants were followed between 1992 and 2012, 156 males and 177 females developed CVD, with risks increasing with increased waist circumference and age. Incidence rates for participants in the 4th waist circumference quartile were 38.3% for males and 47.2% for females [43]. The development of CVD at younger ages has led to recent calls for risk assessment for Aboriginal and Torres Strait Islander adults aged 35 years and under that are supported by the findings from this study [44].

The GPAQ has been demonstrated to be an effective means of collecting data on sedentary and physical activity levels in multiple international communities [2]. It is important to reflect on the appropriateness of the GPAQ for Aboriginal people. Although some aspects of the GPAQ were initially confusing to older participants, any confusion was resolved through explanation from the researcher enabling completion of the questionnaire. Some participants commented on the format of the questionnaire, with too much text on each page. The GPAQ questionnaire was modified to include extra spacing between lines and enlarged font size. This resulted in increased the readability of the GPAQ questionnaire (see Appendix A). Participants became familiar with the GPAQ layout and were able to continue with minimal assistance. This was in contrast to Marshall [15], who described that overall there seemed to be a large degree of confusion amongst the participants when answering the IPAQ-L. Participants had difficulty understanding that the questionnaire and its questions, were broken down into specific categories and considered the examples that the IPAQ-L form uses, such as chopping wood and shovelling snow, were not appropriate for urban Australian Indigenous peoples. One must also consider the method in which the questionnaire was administered, and whether it was delivered in a timely forum that was culturally appropriate [14]. The consultation phase with the Perth Aboriginal community was vital to determine the relevance of the GPAQ and resulted in the inclusion of cultural examples of activities such as hunting and fishing. Another important part of the procedure was having the Noongar first author present while participants completed the questionnaire, to assist understanding and enable cultural safety. Physical activity data from the Australian Bureau of Statistics [36] indicates that physical activity levels have not dramatically changed within the last few years therefore we anticipate the study findings are relevant to the present day. In Perth, lockdown measures that may have impacted physical activity levels were in place for less than two months and were not overly strict e.g., people could exercise outside for unlimited amounts of time. 

## 5. Limitations

There are important limitations related to this preliminary study. Firstly, while the relatively small sample did enable exploration of differences between sub-populations within the sample, elucidation of those differences will require further studies with larger sample sizes. It is also recognized that there is a degree of error associated with self-report of height and weight as used in the study, however community feedback suggested that self-reported height and weight was more culturally appropriate and acceptable. Weight is typically underestimated by approximately 3 kg and height is overestimated by approximately 1–2 cm but usually not sufficient to change the BMI [45]. Finally, the assessment of undiagnosed and undermanaged health conditions relevant to MetS was beyond the scope of this study. 

## 6. Conclusions

The current study is the first to report the use of the GPAQ on the levels and types of physical activity and sedentary time in an Indigenous population in Australia, on Whadjuk, Noongar Boodjar (country) Perth. The findings that physical activity levels were low and sedentary time in this population was high may have contributed to the high prevalence of chronic diseases. These insights will inform the development of health-promotion programmes and strategic interventions to improve and sustain physical activity levels and reduce sedentary time in this population. While there are limitations with self-reported physical activity measurement tools including the GPAQ, this instrument has established reliability and validity in different countries and contexts. This study has demonstrated its applicability in one specific Aboriginal population, with adaptation. Further research including significantly larger sample sizes and Aboriginal communities in a greater range of locations is required to elucidate the gender and age differences suggested in this preliminary study. In addition, further enquiry into culturally appropriate ways to collect data related to height, weight, medication use and self-reported health conditions in Aboriginal communities is warranted.

More robust data on physical activity levels enables community-placed strategies to be developed and implemented, addressing the need to increase physical activity levels and improve health. This has the potential to contribute improvements in the quality of life of Noongar and other Indigenous communities.

## Figures and Tables

**Table 1 ijerph-18-05969-t001:** Clinical characteristics of the participants.

	Male (*N* = 56)		Female (*N* = 73)		All (*N* = 129)		*p* (Male vs. Female)
	Mean	SD	Mean	SD	Mean	SD	
Age (y)	36.1	15	39.1	13.1	37.8	14	0.228
Height (cm)	173	9.7	163.5	9.1	168	10.5	<0.001
Weight (kg)	90.5	20.4	83.7	21.6	86.8	21.3	0.072
BMI * (kg/m^2^)	30.9	8	32.1	8.3	31.5	8.2	0.41
	*n* (%)		*n* (%)		*n* (%)		
Hypertension	10 (18)		24 (33)		34 (26)		0.07
Type 2 diabetes	13 (23)		22 (30)		35 (27)		0.429
High cholesterol	13 (23)		19 (26)		32 (25)		0.838
Heart disease	3 (5)		7 (10)		10 (8)		0.512
Metabolic syndrome	11 (20)		19 (26)		30 (23)		0.529
BMI * (kg/m^2^)							

*: Body mass index.

**Table 2 ijerph-18-05969-t002:** Physical activity (minutes per day) stratified by type of activity and age category. Data are median (interquartile range).

Activity at Work	18–25 (Years, *n* = 25)	26–45 (Years, *n* = 60)	45+ (Years, *n* = 44)	*p* *
Vigorous	0 (0,1)	0 (0,0)	0 (0,0)	0.857
Moderate	0 (0,0)	0 (0,15)	0 (0,0)	0.883
Travel to and from places	0 (0,11)	0 (0,13)	0 (0,13)	0.858
Recreation activities				
Vigorous	0 (0, 18)	0 (0,26)	0 (0,4)	0.029
Moderate	0 (0,14)	0 (0,17)	0 (0,17)	0.453
All activity (work and travel)	29 (5,148)	32 (0,82)	21 (7,60)	0.31
Sedentary	200 (78, 435)	240 (120, 420)	180 (60,300)	0.973
* Kruskal–Wallis test of difference in median values			T2D, type two diabetes	

**Table 3 ijerph-18-05969-t003:** Level of activity by gender (METs per week).

Physical Activity Type	*p*	Gender	Median (25th, 75th)
Work vigorous	0.837	Male	0 (0,0)
		Female	0 (0,0)
Work moderate	0.765	Male	0 (0,120)
		Female	0 (0,0)
Recreation vigorous	0.161	Male	0 (0,1440)
		Female	0 (0,360)
Recreation moderate	0.029	Male	60 (0,900)
		Female	0 (0,240)
Travel to and from places	0.550	Male	0 (0,280)
		Female	0 (0,360)
Total time—work, travel and recreation	0.071	Male	150 (0,435)
		Female	90 (0,240)

In terms of physical activity at work, the category “male moderate” was the only time identified (at 120 MET-minutes per week), while “work vigorous” categories were zero.

**Table 4 ijerph-18-05969-t004:** Activity volume in minutes stratified by risk factor status and gender.

		*N*	Risk Factor Status		*p*
			Normotensive	Hypertensive	0.644
Sedentary	Males	56	180 (143, 375)	270 (90, 465)	
	Females	73	195 (64, 420)	180 (143, 375)	0.568
	All	129	195 (83, 413)	180 (120, 405)	0.497
Active	Males	56	43 (21, 167)	21 (0, 126)	0.141
	Females	73	21 (1, 58)	23 (0, 87)	0.591
	All	129	30 (10, 80)	21 (0, 88)	0.526
		*N*	Nondiabetic	T2D	*p*
Sedentary	Males	56	210 (113, 413)	240 (45, 360)	0.717
	Females	73	180 (79, 420)	225 (128, 368)	0.681
	All	129	180 (105, 420)	225 (105, 323)	0.684
Active	Males	567	44 (21, 147)	221 (4, 102)	0.089
	Females	70	19 (0, 62)	26 (3, 71)	0.604
	All	126	30 (4, 85)	25 (5, 71)	0.969
		*N*	Normal Cholesterol	High Cholesterol	*p*
Sedentary	Males	56	240 (113, 413	180 (75, 450)	0.717
	Females	73	180 (75, 450)	210 (120, 525)	0.514
	All	129	225 (109, 390)	180 (90, 510)	0.798
Active	Males	56	44 (21, 163)	21 (0, 74)	0.089
	Females	73	21 (2, 69)	13 (0, 69)	0.604
	All	129	31 (9, 92)	21 (0, 60)	0.13
		*N*	No heart disease	heart disease	*p*
Sedentary	Males	56	210 (105, 40)	300 (0,180)	0.603
	Females	73	180 (113, 420)	150 (120, 240)	0.243
	All	129	195 (101, 420)	180 (120, 300)	0.971
Active	Males	57	41 (4, 69)	0 (0, 0)	0.107
	Females	70	21 (0, 69)	0 (0, 90)	0.702
	All	127	30 (9, 86)	0 (0, 26)	0.037
Median (25th,75th percentile)					

**Table 5 ijerph-18-05969-t005:** Predictors of physical activity.

			95% Confidence Interval			
Parameter	B	SE	Lower	Upper	Wald	*p*
Intensity = low	0.28	0.32	−0.35	0.91	0.76	0.383
Intensity = moderate	1.39	0.35	0.71	2.07	16.2	<0.001
Age group = 18 to 25 years	0.02	0.47	−0.92	0.95	0	0.973
Age group = 26 to 45 years	0.02	0.38	−0.72	0.76	0	0.959
Age group = > 45 years	0a					
Gender = Males	0.82	0.34	0.16	1.49	5.91	0.015
Gender = Females	0a					
Dependent variable: intensity of physical activity model, age group and gender.						
A, Set to zero because this parameter is redundant; b, fixed at the displayed value.						

## Data Availability

As per the confidentiality as required by Ethic Committee, particularly around the sensitivity around data collect from Indigenous communities the data cannot be shared.

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
