# Peer review of "Physical Activity and Self-Reported Metabolic Syndrome Risk Factors in the Aboriginal Population in Perth, Australia, Measured Using an Adaptation of the Global Physical Activity Questionnaire (GPAQ)"

_ijerph, 2021, doi:10.3390/ijerph18115969_

Round 1
Reviewer 1 Report
Thank you for the opportunity to review this paper on physical activity and morbidity within the Aboriginal population within Western Australia.
General comments
The paper is valuable in that it reports the physical activity of a group which is subject to a number of poor health outcomes. The authors have rightly considered the appropriateness of using existing questionnaires to capture physical activity data for this specific group. I believe it is useful to measure and report on activity levels in this group and use this information to feed into policy, however I feel that the research questions are rather weak. The link between physical activity levels and the chosen health outcomes are well-established in the literature. It is unlikely these associations would be different for this group. I am concerned about the validity of the statistical analysis. The overall sample size is n=129 and a comparison of a number of sub-groups, health outcomes etc. produces some very small comparator groups. It is thus not surprising that results were often non-significant (e.g. in table 5). I believe the authors should think more carefully about the originality of their paper (beyond that of a population perhaps previously under-studied). There is an opportunity to go into further detail and discuss motivators for and barriers to physical activity within this specific group. This could then feed well into policies and legislation to target sedentary behaviour and the resultant morbidity (and likely mortality). Additionally the paper lacks some detail (see comments below) and needs a thorough proof read for typos. I feel the paper is not publishable in its current form.
Abstract (p1)
Should include briefly what metabolic syndrome is. The abstract could be strengthened (and thus point to a more unique aspect of your study) with inclusion of the importance of adapted PA questionnaires for specific populations.
- Introduction
P2, Line 54 – please delete the comma between ‘with’ and ‘lifestyle’
P2, Line 57 – need to include what CVD stands for.
P3, Line 108 – replace ‘he’ with ‘the’
2.2. Participants
Did the first author have particular criteria for those invited to participate e.g. did participants need to be employed (as data on exercise when working is included)? If so this information should be included here. Can the authors comment on the representativeness of the sample’s characteristics in comparison to characteristics of the general population?
2.3. Measures
How was accuracy of the self-reporting of health conditions assessed? Were participants asked if they had previously been diagnosed with high BP, T2 diabetes etc.? Were they asked whether they were taking medication for health conditions? This information should be included here. A potential caveat could be undiagnosed conditions or errors in self-reported height/weight. It would be useful to include the potential for error in self-reported measures within a study weaknesses section within the discussion.
- Results
P5, Line 201 – please include the age range (minimum and maximum).
Table 2 – please include the number in each age group within this table.
I believe the text ‘T2D – type two diabetes’ should be deleted.
Table 4 – there appear to be 57 males and 70 females in table 4 but in table 1 there appear to be 56 males and 73 females – please check/correct these numbers.
- Discussion
Carefully check for/correct typing errors.
The authors should include potential limitations of the study, such as, small sample size (for sub-group analysis) and limitations of using self-reported data.
Author Response
|
Reviewer 1 |
Author Response |
Changes made |
|
|
General comments |
The paper is valuable in that it reports the physical activity of a group which is subject to a number of poor health outcomes. The authors have rightly considered the appropriateness of using existing questionnaires to capture physical activity data for this specific group. I believe it is useful to measure and report on activity levels in this group and use this information to feed into policy, however I feel that the research questions are rather weak. The link between physical activity levels and the chosen health outcomes are well-established in the literature. It is unlikely these associations would be different for this group. I am concerned about the validity of the statistical analysis. The overall sample size is n=129 and a comparison of a number of sub-groups, health outcomes etc. produces some very small comparator groups. It is thus not surprising that results were often non-significant (e.g. in table 5). I believe the authors should think more carefully about the originality of their paper (beyond that of a population perhaps previously under-studied). There is an opportunity to go into further detail and discuss motivators for and barriers to physical activity within this specific group. This could then feed well into policies and legislation to target sedentary behaviour and the resultant morbidity (and likely mortality). Additionally, the paper lacks some detail (see comments below) and needs a thorough proof read for typos. I feel the paper is not publishable in its current form. |
The authors appreciate that the sample size in this stury is only 129 individuals. However, the detailed analysis of GPAQ data in addition to the self-reported MetS data has suggested some important associations between demographic variables and exercise units (MET). Given the scarcity of research using GPAQ with First Nations' peoples, this paper makes an important contribution to the literature. If the GPAQ data had not been disaggregated, important questions about associations between exercise, gender and age would not have been raised. This is entirely consistent with the preliminary nature of this study, in that it indicates directions for further research. (IMPORTANT: include 'preliminary' in the aim and/or introduction section.) |
|
|
Abstract |
Should include briefly what metabolic syndrome is. The abstract could be strengthened (and thus point to a more unique aspect of your study) with inclusion of the importance of adapted PA questionnaires for specific populations. |
The following brief summary was added (2nd sentence of paper) |
MetS is characterised by insulin resistance, abdominal obesity, hypertension, hypertriglyceridemia, high blood-sugar, and low HDL-C |
|
Introduction |
P2, Line 54 – please delete the comma between ‘with’ and ‘lifestyle’ |
cardiovascular disease was inserted in line 59. |
|
|
2.2 Participants |
Did the first author have particular criteria for those invited to participate e.g. did participants need to be employed (as data on exercise when working is included)? If so this information should be included here. Can the authors comment on the representativeness of the sample’s characteristics in comparison to characteristics of the general population? |
No there was no criteria for participant selection. |
|
|
2.3. Measures |
How was accuracy of the self-reporting of health conditions assessed? Were participants asked if they had previously been diagnosed with high BP, T2 diabetes etc.? Were they asked whether they were taking medication for health conditions? This information should be included here. A potential caveat could be undiagnosed conditions or errors in self-reported height/weight. It would be useful to include the potential for error in self-reported measures within a study weaknesses section within the discussion. |
Accuracy of self-reporting was not assessed. Yes, participants were asked if they had been diagnosed with BP, t2D etc. No, no questions were asked in relation to medications prescribed or taken. |
line 401 (discussion section) now reads - A weakness of study is the potential for error in self-reported measures. |
|
3. Results |
P5, Line 201 – please include the age range (minimum and maximum). |
age range has been included (line 206) |
T2D has been removed from the table and replaced with Type 2 Diabetes |
|
4. Discussion |
Carefully check for/correct typing errors. |
The following has been addressed. The authors should include potential limitations of the study, such as, small sample size (for sub-group analysis) and limitations of using self-reported data. |
line 398 (discussion section) now reads - Limitations of study were the potential for error in self-reported measures and small sample size (for sub-group analysis). |
Reviewer 2 Report
Please refer to the attached document for a general and specific comment regarding this manuscript.

Author Response
|
Reviewer 2 |
Author Response |
Changes made |
|
|
A brief summary (one short paragraph) outlining the aim of the paper and its main contributions. As stated in lines 22-25, this study aimed to: 1) establish physical activity levels, including domains and intensity and sedentary behaviour, in the Perth Aboriginal (predominately Noongar) community, and 2) to demonstrate associations between physical activity and MetS risk factors. It is highly relevant that authors report the initial hypothesis and where appropriate to accept or reject it. The manuscript is well written with regards to language, although it may need some minor improvements throughout. The topic is interesting and relevant to improve health related issues in this specific population, although this paper showed major concerns with regards to hypothesis and methods specifically. Most importantly, there is no clear description of the methods and design which makes very hard to understand how some of the main variables were obtained and the points raised by the authors. Lastly, the discussion is in general well conducted, although in the first paragraph it doesn’t present main findings. |
We have added a summary of the main findings to the start of the discussion |
||
|
Abstract |
Line 24: please add space after comma. community, and to… |
Physical activity data from the Australian Bureau of Statistics indicates that physical activity levels have not dramatically changed within the last few years therefore we anticipate the study findings are relevant to present day. In Perth, lockdown measures that may have impacted physical activity levels were in place for less than two months and were not overly strict e.g. people could exercise outside for unlimited amounts of time. COULD ADD THIS AS LIMITATION/STRENGTH NEAR END OF DISCUSSION |
Line 393 now reads -Physical activity data from the Australian Bureau of Statistics indicates that physical activity levels have not dramatically changed within the last few years therefore we anticipate the study findings are relevant to present day. In Perth, lockdown measures that may have impacted physical activity levels were in place for less than two months and were not overly strict e.g. people could exercise outside for unlimited amounts of time. |
|
The introduction is coherent and responds from the general to specific, and from the description of physical activity and inactivity, risk factors, aboriginal population and description of the questionnaires to understand the study variables. |
|||
|
Line 45. Please stay in the same paragraph. |
The paragraph and been added to previous one |
||
|
Introduction |
Line 49. Same as previous. This should be in 1st paragraph not separate. |
The paragraph and been added to previous one |
|
|
Lines 55-58. Please merge both sentences and keep them short. |
Line 57 - The sentence now reads - Aboriginal people are the oldest continuous civilisation in the world have high rates of MetS and CVD. The influence of negative historical and social experiences since the time of colonisation is attributed |
||
|
Line 60. Please reword peoples to population or similar. |
Line 63 - now reads as: ...quality of life of Aboriginal and Torres Strait Islander populations by addressing |
||
|
Line 63. Please reword. These data, as it is repetitive from previous sentence. |
This is the first occasion of life expectancy and hospitalisation is mentioned, |
||
|
Line 72. Please reword 'measures physical activity'. To my understanding, GPAQ doesn’t measure physical activity, this is a qualitative assessment of levels of PA. |
GPAQ is indeed a quantitative measure of physical activity |
||
|
Line 82. Please change to: The aforementioned study concluded. |
Changed |
Line 83 now reads: The aforementioned study concluded study concluded …. |
|
|
Line 85. Please change the word 'measure' to levels. |
The literature consistently uses the word measure and not level so no changes have been made |
||
|
Line 85. Please change the word 'measure' to levels. |
The literature consistently uses the word measure and not level so no changes have been made |
||
|
Line 88. Please stay in the same paragraph. No need to separate from previous. |
New paragraph removed |
||
|
Lines 93-98. What's the rationale for conducting the present study? What will add to the existing evidence? How this helps to improve previous studies and findings? This needs to be clearly reported before authors mention the study aims. Also, is there any reason why you don’t report your main hypothesis? |
GPAQ does indeed measure physical activity, as a valid form of self-reported measurement as described in the GPAQ validiy paper (Bull et al 2009 Global Physical Activity Questionnaire (GPAQ): Nine Country Reliability and Validity Study) e.g. "Measurement of physical activity in large population groups is usually undertaken using self-reported recall, often in the form of a questionnaire conducted either by telephone or household interview" and "This paper summarizes the development of the Global Physical Activity Questionnaire (GPAQ) and the methods protocols and results of an international research collaboration undertaken between 2003 to 2005 to test the measurement properties of GPAQ" |
Can add clarification in the methods that this is a descriptive study. Hence no hypothesis |
|
|
Methods Section |
Line 107. Please revise: 'processes across he study'... |
1. GPAQ is validated,and the Noongar study questionnaire underwent reliability testing and face validity testing? - Describing self-report measures as measures is fine (refer to GPAQ validation again). 2. - Give an example of METS e.g.: One metabolic equivalent (MET) is defined as the amount of oxygen consumed while sitting at rest and is equal to 3.5 ml O2 per kg body weightx min https://onlinelibrary.wiley.com/doi/pdf/10.1002/clc.4960130809 3. Add that self-reported data can be a limitation due to poor or biased recall but that the GPAQ has good validity and is used in many diverse populations around the world |
|
|
Lines 110-113: is this a scientifically validated questionnaire? If so, it could be referenced where and when it was validated. |
line 110 now reads: The development, method,and survey distribution of the questionnaire for the broader investigation within which this study was nested in has been described, validated and published previously and lists reference 15 |
||
|
Statistical Analysis |
Lines 149-150. How do authors justify this assumption? Any reference for this? |
Include the METS reference here, could also refer to the GPAQ analysis guide |
The paper includes two references. |
|
Results Section |
Please consider a figure to report some of your findings as this will add more value to report and visually understand main findings. |
1. I would argue that tables are fine and that figures would not add anything 2. Include the BMI calculation kg/m2 to the methods, mesures section 3. Would also argue that weight is an acceptable term 4. I think Table 1 is fine as it is, thoughts Kristen. Not sure what information they want added to the table, there is a title and definition of abbreviations 5. add to the data analysis section in the methods that statistical significance was set at 0.05 then remove it from line 249. |
|
|
Discussion Section |
The discussion is consistent with the results, however in the first paragraph would be important to point out the two objectives of the study and then point out the main findings of the study. |
Include the main physical activity findings here |
|
|
Lines 262-266. This sentence is too long and confusing. Whilst some parts need to be modified. Please change peoples to populations. |
modified |
Lines 263 to 265 now reads as: This study has demonstrated that the internationally validated GPAQ instrument, culturally adapted for use with the Perth Aboriginal community. It is an acceptable and valid tool to measure physical activity and sedentary behaviour in this context. |
|
|
Line 262. Please change the word measure to assess or similar. Change and revise throughout. |
Measure is correct word |
||
|
Line 271. What do authors exactly mean with 'First Nations'. Please specify. |
(Indigenous) has need inserted in line 268 |
First Nations peoples (Indigenous) from countries with similar settler-colonial |
|
|
Line 277. Please clarify what CVD stands for. |
cardiovascular disease (CVD) |
Line 58 now reads ...as:of MetS and cardiovascular disease (CVD). |
|
|
Lines 283-286. This sentence is confusing and also too speculative. Please revise and keep it in line with the study aims. |
Neither the authors or any reviewers found these lines too speculative and to modify if further information is provided. |
||
|
Lines 296-298. Same as previous, the sentence is speculative and confusing as it stands. |
Neither the authors or any reviewers found these lines too speculative and to modify if further information is provided. |
||
|
Line 299. Add ‘that’ after reported. |
Done |
||
|
Line 320. Please reference that 'Many previous studies... |
Referenced |
||
|
Line 323. Please clarify. 'These data' does it means the ones of the present study or other previous research? |
Data found, in 2012-13, that Aboriginal adults |
Line 326 -Now reads as Data found, in 2012-13, that Aboriginal adults ' |
|
|
Lines 338-339. Please star the new paragraph with a reference to the change of topic. In terms of BMI, authors previously reported a strong link between... Please revise. In addition, the idea of what authors are trying to state is not clear at all. |
Not sure what is unclear |
||
|
Line 358. Please stay in the same paragraph. |
Done |
||
|
Lines 364-365. Please reword 'Incidence rates for participants in the 4th waist circumference quartile'. |
Why? |
||
|
Reword to 4th quartile with regards to waist circumference… |
not sure what reviewer means |
||
|
Lines 391. Please add limitations paragraph before conclusions |
Include the limitations of self-reported questionnaires |
Reviewer 3 Report
The article is interesting and innovative and well thought out. There are just a few small questions for improvement.
How reliable is the self-report of high blood pressure?
It is not well indicated in the measures, variables such as High Cholesterol, Heart Disease, Metabolic Syndrome in table 1.
In table 2 I don't understand how there are so many values that give 0, is it necessary to include some decimals?
The discussion has few quotes that have been used in the introduction.
Round 2
Reviewer 1 Report
Thank you for editing the manuscript in light of my comments. I have two further comments.
There is a still an issue with numbers in your tables. Table one provides characteristics of the full sample (n=129, i.e. 56 males and 73 females) but in table four males=57 (one extra male not included in full sample doesn't make sense). I understand that not all females answered all the questions relating to table 4 but there is a definite error here for males - please fix, thanks.
Page 2, line 86 - please fix typo (repeated words) "The aforementioned study concluded study concluded that more work...".
Author Response
First and foremost, we wish to reiterate our thanks to all reviewers for their valuable suggestions contributing to the strengthening of this paper.
Whilst there may be discipline specific conventions that underpin the use of terms such as weight/mass and estimates/measures we have chosen to use the terms commonly used in descriptive public health studies such as this'. We hope this is acceptable to the reviewers.
The uploaded revised manuscript, together with a detailed response to reviewers, indicates the changes made according to the remarks of each Reviewer.
I look forward to hear from you in the near future
Reviewer 2 Report
Please refer to the attached file.
